# Peripheral endothelial function can be improved by daily consumption of water containing over 7 ppm of dissolved hydrogen: A randomized controlled trial

**Toru Ishibashi**[1,2,3]*, **Kosuke Kawamoto**[1], **Kasumi Matsuno**[1], **Genki Ishihara**[1], **Takamichi Baba**[2], **Nobuaki Komori**[1]

**1** Anicom Speciality Medical Institute, Shinjuku-ku, Tokyo, Japan, **2** Department of Rheumatology, Orthopaedic Surgery and Health Care, Huis Ten Bosch Satellite H2 Clinic Hakata, Hakata-ku, Fukuoka, Japan, **3** Department of Orthopaedic Surgery, Yonemori Hospital, Kagoshima-shi, Kagochima, Japan

* tishibashi@h2-lab.jp

**Data Availability Statement:** All relevant data are within the paper and its Supporting Information files.

## Abstract

### Background

Measurement of the reactive hyperemia index (RHI) using peripheral arterial tonometry (PAT) has shown benefits in the evaluation of vascular endothelial function and prediction of cardiovascular disease prognosis. Thus, it is important to examine the factors that promote the RHI. In this study, we aimed to investigate the effect of molecular hydrogen ($H_2$) on reactive hyperemia-PAT of the small arteries of fingers in healthy people.

### Methods

To determine the efficacy of $H_2$ for improving peripheral vascular endothelial function, water containing high $H_2$ concentrations was administered to participants, and the Ln_RHI was measured in the finger vasculature. Sixty-eight volunteers were randomly divided into two groups: a placebo group (n = 34) that drank molecular nitrogen ($N_2$)-containing water and a high $H_2$ group (n = 34) that drank high $H_2$ water (containing 7 ppm of $H_2$: 3.5 mg $H_2$ in 500-mL water). The Ln_RHI was measured before ingesting the placebo or high $H_2$ water, 1 h and 24 h after the first ingestion, and 14 days after daily ingestion of high $H_2$ water or the placebo. The mixed effects model for repeated measures was used in data analysis.

### Results

The high $H_2$ group had a significantly greater improvement in Ln_RHI than the placebo group. Ln_RHI improved by 22.2% ($p<0.05$) at 24 h after the first ingestion of high $H_2$ water and by 25.4% ($p<0.05$) after the daily consumption of high $H_2$ water for 2 weeks.

**Funding:** The authors received no specific funding for this work.

**Competing interests:** The authors have declared that no competing interests exist.

## Conclusions

Daily consumption of high $H_2$ water improved the endothelial function of the arteries or arterioles assessed by the PAT test. The results suggest that the continuous consumption of high $H_2$ water contributes to improved cardiovascular health.

## Introduction

Vasomotor tone and appropriate blood pressure are maintained by the vascular endothelium, which responds to shear stress generated by increased blood flow and stimulates vasodilation [1]. Vascular damage is caused and intensified by endothelial dysfunction, which induces chronic inflammation of the vasculature, followed by the development of atherosclerosis and cardiovascular disease [2–6]. The endothelium is composed of a monolayer of endothelial cells that function as an endocrine organ and a 0.2-2-μm-thick, negatively charged surface layer called the glycocalyx, which is a primary-functional barrier for endothelial cell protection [7, 8]. The endothelium senses the shear stress and transduces it into chemical or electrical signals, including nitric oxide (NO) and endothelium-derived hyperpolarization factor (EDHF), leading to the relaxation of smooth muscle cells. In smaller diameter arteries, such as peripheral arteries, EDHF predominantly controls the vasomotor function, while in larger diameter vessels, such as conduit arteries, NO plays a pivotal role in the vascular response to blood flow. These factors do not work alone but work in conjunction with each other to maintain a healthy circulation. The method to monitor the response of a conduit artery endothelium is flow-mediated dilation (FMD) [9], and reactive hyperemia-peripheral arterial tonometry (RH-PAT) has been developed to assess endothelial function in the small arteries of the fingers [10]. These methods are thought to significantly predict the risks of cardiovascular and cerebrovascular events, including angina, myocardial infarction, ischemic stroke, cerebral infarction, acute coronary syndrome, and heart failure [2–5]. Recently, RH-PAT and FMD have been compared, and it was suggested that these two methodologies are not associated and reflect different aspects of cardiovascular risks. [10–13]. It is important to not only estimate the function of the endothelium of both conduit arteries and peripheral arteries in healthy people using these non-invasive methodologies, but to also identify the factors or methodologies that can improve endothelial function and prevent endothelial dysfunction in healthy people without administering pharmaceutical drugs.

Recently, we showed that consumption of over 3.5 mg of molecular hydrogen ($H_2$) dissolved in water (at a concentration of 7 ppm $H_2$ in 500 ml of water: high $H_2$ water) can rapidly improve FMD of the conduit brachial artery [14]. In this study, we aimed to investigate the effect of $H_2$ on RH-PAT of the small arteries of fingers in healthy people. In some interventional studies, attempts to improve the reactive hyperemia index (RHI) without using a pharmaceutical approach, but instead focusing on exercise therapy and/or lifestyle improvement, have been reported in patients suffering from heart failure and metabolic syndrome [15–18]. Improvements in RHI were also reported when postprandial hyperglycemia was treated pharmaceutically [19]. Besides these disease conditions, it is important to identify safe and conventional ways for achieving relatively high (healthy) RHIs in the small vessels of healthy individuals. We demonstrated here whether daily consumption of high $H_2$ water is beneficial for the endothelial function of which improvement can prevent the development of atherosclerosis and cardiovascular disease.

## Methods

### Subjects and the measurements of RHI

This study was designed to assess the possibility of $H_2$ acting as an agonist for NO and/or EDHF, which was suggested previously using FMD (S1 File). The research protocol was registered with the UMIN clinical trial registry (number: UMIN000032510; dated 08/05/2018). Our study was approved by the Huis Ten Bosch Satellite H2 Clinic Ethics Committee (July 3, 2015; approval number: hshc1502). The study was performed in accordance with the Declaration of Helsinki. Seventy-one volunteers were recruited between July 2015 and May 2018 and informed individually about the significance of the study and the instruments of intended measurement. These volunteers enrolled in this study after providing their written informed consent. This clinical trial was performed according to the S1 Checklist, and the CONSORT diagram is shown in Fig 1. All study subjects took no medications or dietary supplements or received any medical treatments for more than 6 months before this study was performed. All subjects were asked to fast and avoid drinking caffeinated drinks or ingesting sugar as well as breakfast for 6 h before the test. The subjects were randomly divided into two groups, and three of them did not complete the protocol. Randomization was performed by blinded independent physicians' assistants. Sixty-eight subjects completed the study protocol, and the average (±standard deviation [SD]) age of the 34 subjects (men: 13; women: 21) in the high $H_2$ group was 36.3±10.4 years and that of the 34 subjects (men: 11; women: 23) in the placebo group was 38.2±11.2 years. Using the EndPad2000 System (Itamar Medical Inc.), we measured the RHI using endothelium-dependent digital pulse amplitude testing (EndoPAT) according to the reports in previous studies [2,18]. In brief, disposable RH-PAT probes were placed individually on both index fingers, and a blood pressure cuff was placed on the subject's ipsilateral upper arm region in a quiet and dimmed room. After resting for more than 20 min, a baseline pulse amplitude recording was started, and RH-PAT was induced via reperfusion of blood flow after a 5-min cuff occlusion of the brachial artery at 60 mmHg above the measured systolic pressure. Next, the subjects drank 500 ml of the placebo water or the high $H_2$ water within 10 min, and after resting for 1 h, the RHI was measured again. On the second day, 24 h after the first ingestion of placebo or high $H_2$ water and before the second ingestion, the RHI was measured. After the measurements were obtained, a second drink was administered, and the day after, they drank the placebo or the high $H_2$ water once each day until the day before the last measurements at 14 days. In total, they drank the high $H_2$ water or placebo 14 times. No adverse events were recorded as a result of the clinical study. The RHI was calculated as follows: RHI = Ln{[RH-PAT ratio]×[0.226×Ln (baseline)−0.2]} [2, 3, 20], and we used the natural logarithmic scaled Ln_RHI.

### Preparation of high $H_2$ water and placebo water

The high $H_2$ water (Aquela Global Co., Ltd., London, UK) was prepared according to previously described methods [21]. Briefly, $H_2$ gas was produced in an elastic polyethylene terephthalate (PET) bottle (manufactured by TOMIKAWA Chemical Industry Co., Ltd., or other bottles were used if they were prepared for use with carbonated drinks and had a high elasticity to withstand the high pressures of the dissolved gas). The bottle was filled with 530 ml of water by mixing 75% of metal aluminium grains with 25% of calcium hydroxide (by weight) and 0.5 ml water in an acrylic resin tube placed into the bottle. During the reaction, the $H_2$ gas pressed against the surface of the water in the standing bottle, which was gradually hardened by the increasing pressure of the emerging hydrogen gas. After the reaction was terminated, the $H_2$ gas was dissolved in the water for 12 h, and the bottle was shaken for about 30 s just before

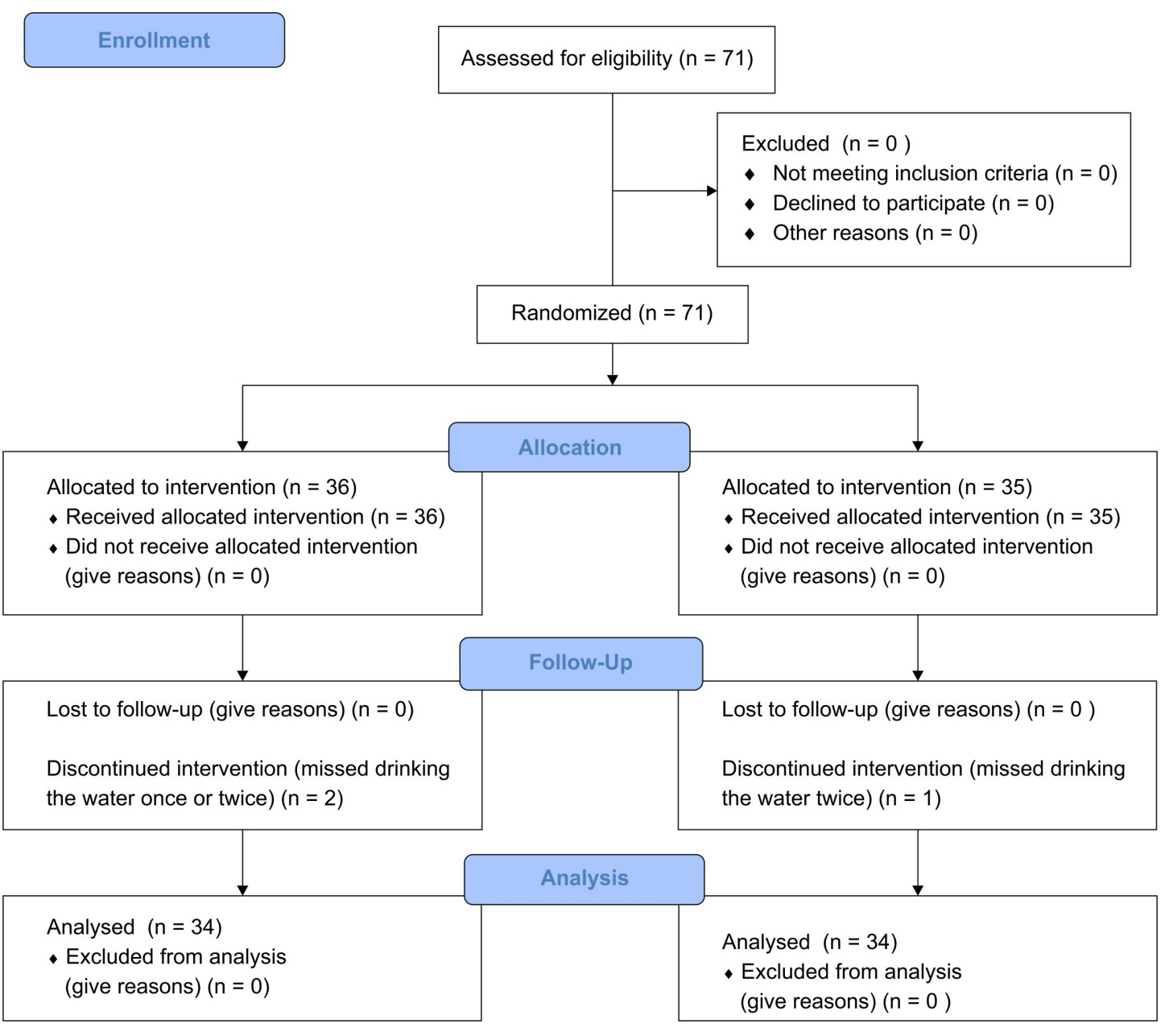

**Fig 1. CONSORT flow diagram.**

drinking. The placebo water was prepared by filling up a PET bottle containing water with molecular nitrogen ($N_2$) gas under 0.8 MPa to make the placebo bottles as firm as the bottles with the high $H_2$ water. An acrylic resin tube with a placebo non-woven fabric (used to produce $H_2$ gas in a tube but lacking the reactive compound) was also placed in the placebo bottles so that the volunteers could not distinguish between the two types of water. On the first and second days of the EndoPAT testing, participants were given either a placebo or high $H_2$ PET bottle of water, which was indistinguishable due to the lack of taste and identical appearance. After the second day, each PET bottle of water was shipped to the participants daily, with each bottle received a day before the administration. It has been clarified and agreed in this field

that $H_2$ is transferred into the human body and circulated from ingested water containing $H_2$. In cases where rats were administered 15 mL/kg of 0.8 mM $H_2$ water by a catheter, or mice drank approximately 200 mL/kg of 0.2–0.4 mM $H_2$ water per day, the concentration of $H_2$ in their blood was 5–6 μM and about 2 μM, respectively [22, 23]. In a clinical report about the concentration of $H_2$ in blood 30 minutes after ingesting water containing $H_2$, we also clarified that the presence of hydrogen gas during exhalation from the lung of the person who ingested the high $H_2$ water can be explained by the rapid and passive diffusion of $H_2$ in the aqueous human body including circulation [21].

## Statistical analysis

The efficacy of the high $H_2$ water was estimated by analyzing the change in Ln_RHI at each time point based on the value of Ln_RHI before drinking, which reflected the change in Ln_RHI from baseline. We used the mixed effects model for repeated measures (MMRM) for the analysis. MMRM is a model used to analyze repeated measures continuous outcomes [24]. Using all data obtained at 1 h, 24 h, and 2 weeks, the MMRM included the treatment group, time points, and the interaction between the treatment group and the time points as fixed effects, with the Ln_RHI at baseline as a covariate. In the MMRM analysis, the correlation structure between time points needs to be specified and unstructured (i.e., no specific correlation structure), which is why we used it. Furthermore, we assumed a Gaussian error term in the model. We performed a log-likelihood ratio test and residual analysis in order to check the fit of the model. Under the MMRM, the adjusted mean change from the baseline, adjusted mean difference between the treatment groups, and their 95% confidence intervals (CIs) and $p$-values were calculated using the software SAS version 9.2 (SAS Institute, Inc. Cary, NC, USA).

## Results

The high $H_2$ water was prepared more than 12 h before the subjects drank it, and we confirmed that the concentration was over 7 ppm, as described before [21]. The participants drank about 500 ml of the water containing 7-ppm $H_2$ immediately after opening the PET bottle so that they could consume approximately 3.5 mg $H_2$.

The baseline data of the subjects are presented in Table 1. All of the subjects had normal mean biometric parameters and had no previous diagnosis of, or took medications related to metabolic syndrome, hypertension, chronic inflammatory diseases, or cardiovascular diseases.

**Table 1. Characteristics of the enrolled volunteers.**

|  | All cases (N = 68) | High $H_2$ group (n = 34) | Placebo group (n = 34) |
|---|---|---|---|
| **Age** | 37.3±10.7 | 36.3±10.4 | 38.2±11.2 |
| **Men, n (%)** | 24 (35.3) | 13 (38.2) | 11 (32.3) |
| **Women, n (%)** | 44 (64.7) | 21 (61.8) | 23 (67.6) |
| **Height (m)** | 1.61±0.09 | 1.62±0.08 | 1.60±0.09 |
| **Weight (kg)** | 56.4±11.0 | 57.4±11.1 | 55.3±10.7 |
| **BMI (kg/m²)** | 21.6±3.34 | 21.7±3.84 | 21.5±2.81 |
| **SBP (mmHg)** | 114±14.7 | 113±15.4 | 115±14.2 |
| **DBP (mmHg)** | 67.5±10.9 | 66.9±12.1 | 68.0±9.76 |
| **HR (beats/minute)** | 71.0±8.75 | 70.7±8.13 | 71.4±9.44 |

BMI: body mass index, SBP: systolic blood pressure, DBP: diastolic blood pressure, HR: heart rate, $H_2$: hydrogen. The data are presented as mean±standard deviation.

There were no significant differences in age, sex, height, weight, body mass index (BMI), blood pressure, or heart rate between the two groups. Additionally, there was no significant difference in the RHI value of subjects between the groups (placebo group: 0.61±0.26, high $H_2$ group: 0.63±0.29). We considered that the subjects in these two groups had physiologically similar characteristics in terms of endothelial function.

The concentration of $H_2$ in the bloodstream is thought to reach its peak within 10 min [21]. The influence of $H_2$ was observed with FMD of the brachial artery at 30 min after $H_2$ consumption in a previous study [18]. Thus, we obtained the first measurement of Ln_RHI at 1 h after the subjects drank the high $H_2$ water so that the endothelium of the peripheral artery had enough time to respond to the $H_2$ mechanism, whether or not it was NO dependent, as observed in the study using FMD. The Ln_RHI data "before drinking" was collected 30 min before the first drink of the high $H_2$ water. The interval between the first PAT measurement (before drinking) and the second measurement was 1.5–2.0 h. It has been reported that repetitive PAT measurements over a 1-hour or 2-hour interval do not carry the effects forward [25]. The second measurement was set 24 h after the first drink, but before the second drink on the second day. All of the tests were performed in the afternoon and the subjects drank the placebo or the high $H_2$ water once a day at roughly the same time in the afternoon for 14 days. The third measurement was performed on the 15th day after the subjects finished the daily drinking of the placebo or the high $H_2$ water for 14 days.

As shown in Fig 2, 24 h after the first consumption of 3.5 mg $H_2$ in water, the Ln_RHI improved significantly compared to the placebo water. By continuous consumption of the high $H_2$ water for 2 weeks, Ln_RHI was ameliorated compared to the placebo group. The statistical analysis comparing all of the subjects in the two groups is summarized in Table 2. From the result of log-likelihood ratio test ($p = 0.0483$) and the analysis of residuals (data not shown), we confirmed the appropriateness of the model to the data. The improvement in RH from its level before ingestion was also observed with the $H_2$ group. At 1 h after the first consumption of the high $H_2$ water, the Ln_RHI tended to increase from the value observed before consumption ($p = 0.0597$). At both 24 h and 2 weeks after ingestion, significant improvement was observed in the Ln_RHI, with $p$-values of 0.0003 and 0.0066 for 24 h and 2 weeks,

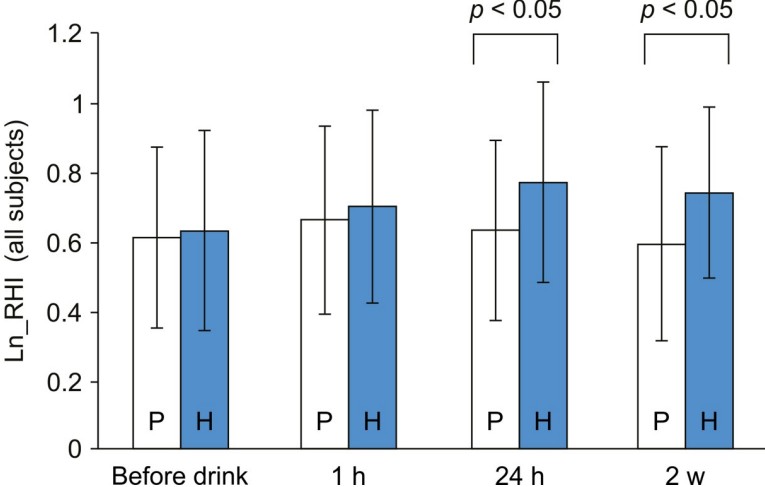

**Fig 2. Ln_RHI values for all of the subjects.** Results are presented as the height of the columns at each time point. P: the placebo group, H: the high $H_2$ group. Error bars are indicated on each column, and $p$-values are indicated above the compared columns between the two groups. RHI: reactive hyperemia index; $H_2$: hydrogen.

**Table 2. Comparison between the two groups for all of the subjects.**

| Time Point | Treatment Group | Change from the Ln_RHI Before Drinking | | | H$_2$ Group Versus Placebo Group | |
|---|---|---|---|---|---|---|
| | | LS Mean (Standard Error) | 95% Confidence Interval | *p*-value | Difference of Adjusted Mean [95% Confidence Interval] | *p*-value |
| **1 h** | Placebo | 0.05 (0.04) | -0.03, 0.13 | 0.1849 | | |
| | H$_2$ | 0.07 (0.04) | 0.00, 0.15 | 0.0597 | 0.02 [-0.09, 0.13] | 0.6851 |
| **24 h** | Placebo | 0.02 (0.04) | -0.06, 0.09 | 0.6456 | | |
| | H$_2$ | 0.14 (0.04) | 0.07, 0.22 | 0.0003 | 0.13 [0.02, 0.23] | 0.0193 |
| **2 weeks** | Placebo | -0.02 (0.04) | -0.11, 0.07 | 0.6395 | | |
| | H$_2$ | 0.12 (0.04) | 0.03, 0.21 | 0.0066 | 0.14 [0.02, 0.26] | 0.0237 |

MMRM analysis: fixed effects are for the treatment groups, time points, and interaction effects.

The covariate was set as the Ln_RHI value before drinking. The covariance structure was unstructured.

RHI: reactive hyperemia index, LS: least square mean, H$_2$: hydrogen.

*p*-value of log-likelihood test: 0.0483.

respectively. By contrast, in the placebo group, no improvement of Ln_RHI was observed throughout the study. The *p*-values are presented in Table 2.

In order to identify the influence of the Ln_RHI value before ingestion on the data at each time point, we divided the subjects into two groups with the cut-off value of 0.71 (see Discussion section), a low Ln_RHI group (<0.71) and a high Ln_RHI group (≥0.71). Nineteen subjects in the high H$_2$ group and 22 subjects in the placebo group were sorted in the low Ln_RHI group. We also applied MMRM analysis to the low Ln_RHI group. The p-value of the log-likelihood ratio test was 0.3597. It was relatively large because of the small sample size. From the result of the analysis of residuals (data not shown), we confirmed the appropriateness of the model to the data. Comparing the two groups, in the low Ln_RHI group, significant improvement was observed only after daily consumption of the high H$_2$ water for 2 weeks, as shown in Fig 3. When the Ln_RHI at each time point was compared to the Ln_RHI before ingestion in the high H$_2$ group, a significant improvement was observed for all of the 3 time points. Significant improvement of Ln_RHI at 1 h (*p* = 0.0057) and 24 h (*p* = 0.0205) compared to the values before ingestion was also seen in the placebo group. The statistical analysis is summarized in Table 3. On the other hand, in the high Ln_RHI group, there was no significant difference between the placebo and the high H$_2$ group, and no significant improvements were observed when the Ln_RHI was compared to the Ln_RHI value before ingestion in either the placebo or the high H$_2$ group (data not shown).

## Discussion

In this study, consumption of 3.5 mg of H$_2$ dissolved in water improved Ln_RHI after daily consumption for 2 weeks (Fig 2). A single drink of the high H$_2$ water also increased the Ln_RHI in the all of the subjects in the high H$_2$ group at 24 h after the drink. It was suggested that daily consumption of high H$_2$ water is important to improve the endothelial function of the peripheral artery. Furthermore, to investigate how the value of Ln_RHI before drinking affects the H$_2$ mode of action, we divided the subjects into two groups, with high risk and with low risk based on Ln_RHI values. Although not intended for healthy subjects, there are several investigations on the prognostic value of Ln_RHI for predicting cardiovascular risks. The Ln_RHI values reported were 0.531 in 577 patients at high risk for cardiovascular events by Matsuzawa et al [2] and 0.4 in 270 low-risk outpatients with chest pain by Rubinshtein et al [3]. It has been also reported that the risk of arrhythmia recurrence could be estimated by an

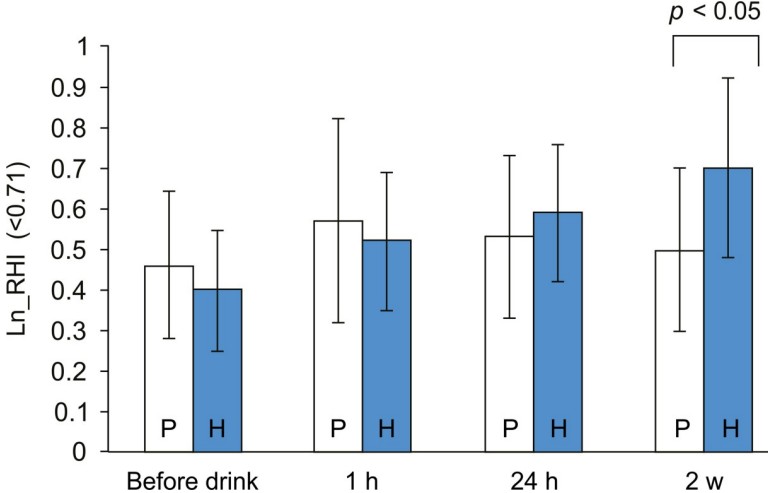

**Fig 3. Ln_RHI values for the low Ln_RHI group (Ln_RHI <0.7).** Results are presented as the height of the columns at each time point. P: the placebo group, H: the high $H_2$ group. Error bars are indicated on each column, and $p$-values are indicated above the compared columns between the two groups. RHI: reactive hyperemia index, $H_2$: hydrogen.

Ln_RHI value of 0.618 in 92 participants with atrial fibrillation undergoing catheter ablation [26]. Orthopedic patients who underwent total hip or knee arthroplasty and avoided postoperative deep vein thrombosis (DVT) had preoperative ln-RHI values of 0.71±0.25 [27]. Among these reports, the border value of 0.71 for the orthopedic patients avoiding DVT seems to be close to that of the healthy masses. To predict future endothelial health in healthy subjects in this study, it seemed to be appropriate to set the threshold for RHI above the values for patients with cardiovascular risks. We set the Ln_RHI value at 0.71 in order to divide the subjects into two groups.

In the low Ln_RHI group, the continuous consumption of $H_2$ for 2 weeks significantly improved the Ln_RHI value compared to placebo. An unexpected result that we found was that there was no improvement at 24 h after the first drink. To account for this result, both the time at 1 h and at 24 h after the single drink, showed significant improvement from the values before ingestion of the drink were also observed within the placebo group (Table 3). Because

**Table 3. Comparison between the two groups with the low Ln_RHI (Ln_RHI <0.71).**

| Time Point | Treatment Group | Change from Baseline | | | $H_2$ Group Versus Placebo Group | |
|---|---|---|---|---|---|---|
| | | LS Mean (Standard Error) | 95% Confidence Interval | $p$-value | Difference of Adjusted Mean [95% Confidence Interval] | p-value |
| **1 h** | Placebo | 0.13 (0.05) | 0.04, 0.22 | 0.0057 | | |
| | $H_2$ | 0.11 (0.05) | 0.01, 0.20 | 0.0349 | -0.03 [-0.16, 0.11] | 0.7011 |
| **24 h** | Placebo | 0.09 (0.04) | 0.01, 0.17 | 0.0205 | | |
| | $H_2$ | 0.17 (0.04) | 0.09, 0.25 | 0.0002 | 0.08 [-0.04, 0.19] | 0.1788 |
| **2 weeks** | Placebo | 0.06 (0.05) | -0.03, 0.16 | 0.1625 | | |
| | $H_2$ | 0.28 (0.05) | 0.18, 0.38 | < .0001 | 0.22 [0.08, 0.35] | 0.0024 |

MMRM analysis: fixed effects are for the treatment group, time points, and interaction effects.

The covariate is set as the Ln_RHI value before drinking. The covariance structure is unstructured.

RHI: reactive hyperemia index, LS: least square mean, $H_2$: hydrogen.

$p$-value of log-likelihood test: 0.3597.

there was no improvement at 2 weeks within the placebo group, the placebo effects are thought to be transient, although it is currently unclear whether it is merely the placebo effect or there is another factor that causes the improvement of Ln_RHI within 1 h after the placebo group drinks the water containing $N_2$. On the other hand, there was no influence of $H_2$ intake on RH-PAT in the high Ln_RHI group. The mechanism in which $H_2$ acts on the peripheral endothelium seems to be mediated by the factors that are lacking or are impaired in the low Ln_RHI group with future risk for endothelial dysfunction. In a recent report, we demonstrated the preliminary potential of $H_2$ and its agonistic effects on NO-related mechanisms, as well as the functional roles of $H_2$ on the endothelium of conduit arteries by the FMD test [14]. Similar to the FMD test, NO-mediated vasodilation mechanisms are certainly involved in PAT testing. It was clearly demonstrated that the PAT test evaluates NO production, using $N^G$-nitro-L-arginine methyl ester (L-NAME) as an inhibitor of nitric oxide synthase [28]. However, when the reactive increase in reperfusion flow in the conduit brachial artery at the site of occlusion in the upper arm is inhibited by L-NAME, the increase in reactive hyperemia and reperfusion flow to the finger vessels seems to decrease. Therefore, it is necessary to consider whether reperfusion flow without the reactive increase is enough to cause the release of EDHF. A prompt release of EDHF may be required in addition to the release of NO to obtain a high enough RHI to maintain the health of the peripheral microcirculation, although there are few studies on EDHF and the RH-PAT test. Furthermore, NO and EDHF are thought to cause vasodilation via mitochondrial function, and endothelial dysfunction is mediated by the overproduction of superoxide and $H_2O_2$ [29].

The accumulated molecular mechanisms of endothelial function regarding the vasodilation of peripheral described in previous reports are summarized and presented in Fig 4 to help considering the mechanistic insights of $H_2$ on PAT data and EDHF [29–36].

As illustrated in the left part of Fig 4, as long as the endothelium functions well with high RHI values (it is above 0.7, here), the peripheral vasculature can metabolize and generate appropriate amounts of energy with relevant $O_2$ consumption. The by-products of mitochondrial electron transfer ($H_2O_2$ induced by flow-mediated shear stress via the function of superoxide dismutase in respiratory conditions) are moderate and play important roles as signals of vasodilation, as evidenced in the coronary arteries [30,31]. The subsequent activation of large conductance $Ca^{2+}$-activated potassium channels in smooth muscle cells induces EDHF and relaxation of the vessels. The secured flow by EDHF maintains the microcirculation, which supports the appropriate metabolism of the endothelial mitochondria. Then, it creates the positive feedback loop where the higher value of RHI is provided.

By contrast, the hypoxic microcirculation which causes an accumulation of NADH or $FADH_2$ and an increase in electron leakage from the electron transport chain, with excess production of superoxide and $H_2O_2$ resulting in the formation of deleterious peroxynitrite ($ONOO^-$) by exhausting NO as well as the formation of hydroxyl radicals [29]. Once the vicious circle presented in the right part of Fig 4 is established, RH in the peripheral arteries including coronary arterioles could be impaired, as in the low Ln_RHI group.

It should be noted that measuring RH of the peripheral microcirculation in which occlusion of the brachial artery following the release of blood flow is known as the model of ischemia reperfusion (I/R) injury, although the function of the vasodilation of the microvasculature is not fully understood [32]. The mechanisms of the cellular damage by I/R injury are believed to be caused by the superoxide burst in the mitochondrial respiratory chain. Particularly, the reverse electron transport accompanied by the elevated mitochondrial membrane potential is the major factor of electron leakage in mitochondrial complex I [33]. Recently, we have reported that $H_2$ reduces the mitochondrial membrane potential in the living cultured cells [34]. In the study, the overproduction of superoxide with the high ratio of $NADH/NAD^+$,

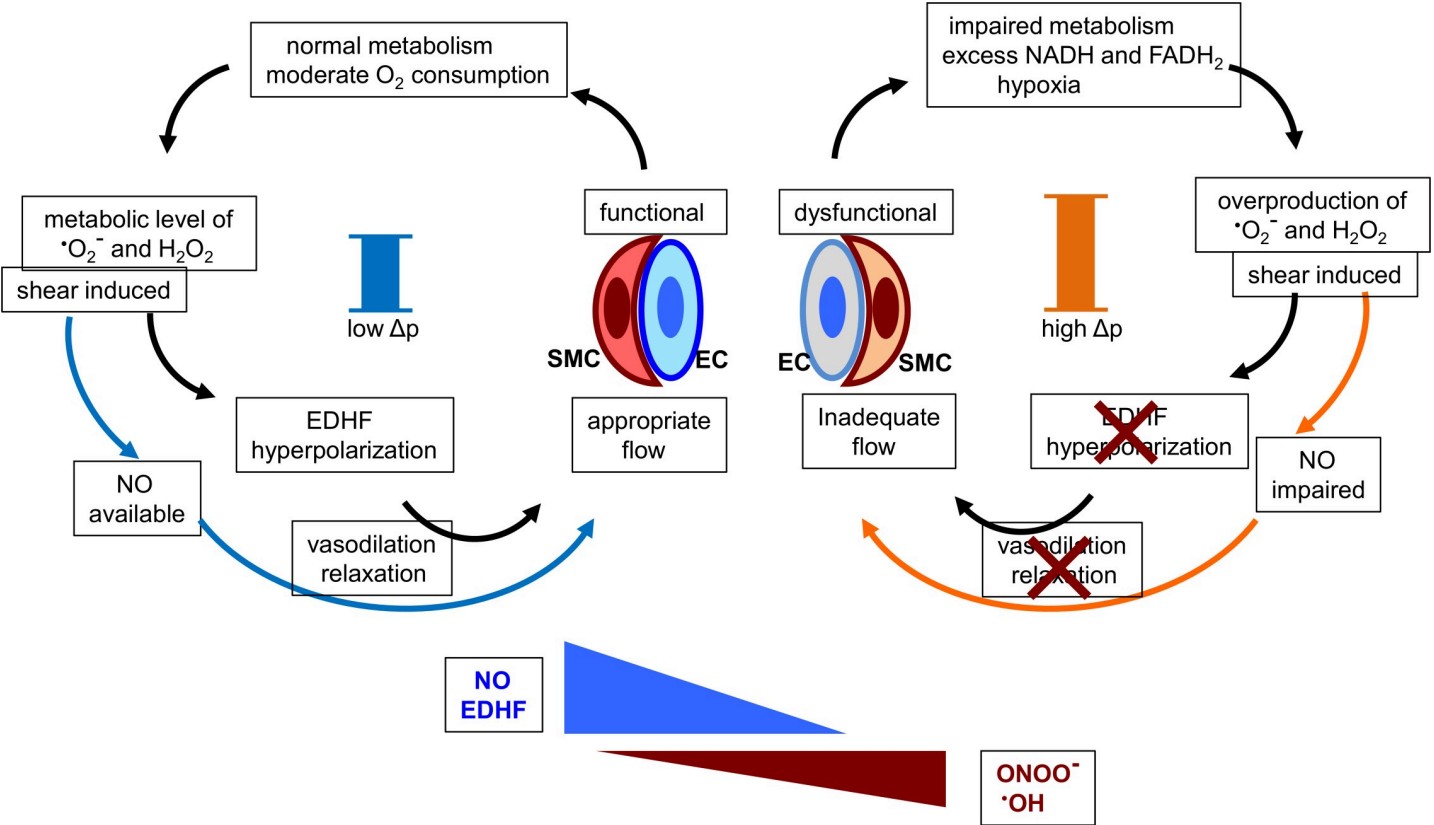

**Fig 4. Schematic representation of the possible action of H₂ for the improvement of RHI.** EC: endothelial cell, SMC: smooth muscle cell. The crosses indicate the impairment of hyperpolarization and/or vasodilation/relaxation, and the bars in the center of both of the circles represent the mitochondrial membrane potential (Δp).

which reflects the highly reduced state of mitochondria during hypoxia following I/R injury [35], was suppressed by $H_2$. This situation corresponds to the right circle in Fig 4. On the basis of these most recent mechanistic insights of $H_2$ [34, 36], we suppose that $H_2$ may improve endothelial function via the mitochondrial mechanisms in healthy subjects. $H_2$ may shift the circle in Fig 4 from right to left by reducing the elevated mitochondrial membrane potential as well as the overproduction of superoxide under the condition with hypoxia presented in the right circle.

Thus far, the limitations to explain the mechanisms of the $H_2$ effect on a wide range of diseases or pathological states based on the scavenging property of $H_2$ with respect to highly reactive radicals have been discussed [37, 38]. In blood flow, the availability of $H_2$ and/or the frequency with which it reacts with target radicals seems insufficient to explain the distinct biological effects of $H_2$ in the peripheral vasculature. In the present study, the continuous scavenging of deleterious radicals by $H_2$ is excluded by the single intake of high $H_2$ water in a day. Therefore, we consider that the effects of $H_2$ herein are occur via the mitochondrial mechanisms described in Fig 4 [33–36].

## Study limitations

As the aim of this study was to investigate the effect of $H_2$ on RH-PAT intended for healthy subjects to assess the significance of daily consumption of $H_2$ to prevent the endothelial dysfunction of small arteries, there are no data for an atherosclerotic group with a low RHI value.

It is important to examine the effect of $H_2$ on the RH-PAT of people with a high risk of hypertension, cardiovascular disease, diabetes mellitus and dyslipidemia due to atherosclerosis. The next study should include participants with these characteristics. As for glucose metabolism and lipid profiles, which are confounding factors for endothelial function, we did not collect blood samples, and our study protocol did not include the blood test. In addition, the number of the subjects in this study was small and the follow-up data for the long-term consumption of high $H_2$ water was lacking.

## Conclusions

The daily consumption of water containing a high concentration of $H_2$ (over 7 ppm or 3.5 mg in 500 mL of water) could ameliorate the endothelial function of the arteries and arterioles assessed by the PAT test. It was suggested that the high $H_2$ water is useful for improving the function of the vasculature and decreasing the risk of illness. Based on the agonistic function of $H_2$ against the endothelium as suggested herein, a controlled study with a larger number of subjects and a long-term observation should be done in the near future. High $H_2$ water has been available commercially, and now many people have been drinking it in Japan and Europe. A cohort study of this population should also be performed in the future.

## Supporting information

**S1 Checklist. CONSORT 2010 checklist of information to include when reporting a randomised trial**[*]**.**
(DOC)

**S1 File. Trial protocol.**
(DOCX)

**S2 File.**
(DOCX)

**S1 Data. Set of the present study.**
(XLSX)

## Acknowledgments

We thank T. Nagao, H. Tagomori, K. Kiyota, R. Nawata, K. Fukuoka, and M. Takeda for the technical support and advice. We would like to thank Editage for English language proofreading.

## Author Contributions

**Conceptualization:** Toru Ishibashi, Kosuke Kawamoto, Nobuaki Komori.

**Data curation:** Kosuke Kawamoto, Kasumi Matsuno.

**Formal analysis:** Toru Ishibashi, Kosuke Kawamoto, Genki Ishihara, Takamichi Baba, Nobuaki Komori.

**Investigation:** Kosuke Kawamoto.

**Writing – original draft:** Toru Ishibashi.

**Writing – review & editing:** Toru Ishibashi, Kosuke Kawamoto, Kasumi Matsuno, Genki Ishihara, Takamichi Baba, Nobuaki Komori.

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
