## [Decision Letter · Decision Letter 0]

28 Nov 2019

PONE-D-19-26280

Peripheral endothelial function can be improved by daily consumption of water containing over 7ppm of dissolved hydrogen: A randomized controlled trial

PLOS ONE

Dear Dr. Ishibashi,

Thank you for submitting your manuscript to PLOS ONE. After careful consideration, we feel that it has merit but does not fully meet PLOS ONE’s publication criteria as it currently stands. Therefore, we invite you to submit a revised version of the manuscript that addresses the points raised during the review process.

Clinical reviewers have raised serious concerns about study protocol, about how this study compares with previous literature, about manuscript presentation, about statistical analysis. All reviewers comments should be seriously taken into consideration and addressed properly. 

We would appreciate receiving your revised manuscript by Jan 12 2020 11:59PM. To enhance the reproducibility of your results, we recommend that if applicable you deposit your laboratory protocols in protocols.io, where a protocol can be assigned its own identifier (DOI) such that it can be cited independently in the future. For instructions see: http://journals.plos.org/plosone/s/submission-guidelines#loc-laboratory-protocols

We look forward to receiving your revised manuscript.

Kind regards,

Giuseppe Andò, M.D., Ph.D.

Academic Editor

PLOS ONE

Journal Requirements:

3. Please note that all PLOS journals ask authors to adhere to our policies for sharing of data and materials: https://journals.plos.org/plosone/s/data-availability. According to PLOS ONE’s Data Availability policy, we require that the minimal dataset underlying results reported in the submission must be made immediately and freely available at the time of publication. As such, please remove any instances of 'unpublished data' or 'data not shown' in your manuscript and replace these with either the relevant data (in the form of additional figures, tables or descriptive text, as appropriate), a citation to where the data can be found, or remove altogether any statements supported by data not presented in the manuscript.

Reviewers' comments:

Reviewer's Responses to Questions

**Comments to the Author**

1. Is the manuscript technically sound, and do the data support the conclusions?

Reviewer #1: No

Reviewer #2: Partly

Reviewer #3: Partly

2. Has the statistical analysis been performed appropriately and rigorously? 

Reviewer #1: Yes

Reviewer #2: Yes

Reviewer #3: No

3. Have the authors made all data underlying the findings in their manuscript fully available?

Reviewer #1: No

Reviewer #2: No

Reviewer #3: Yes

4. Is the manuscript presented in an intelligible fashion and written in standard English?

Reviewer #1: Yes

Reviewer #2: Yes

Reviewer #3: Yes

5. Review Comments to the Author

Reviewer #1: In this manuscript, Ishibashi et al. describe the effects of hydrogen water for 2 weeks on endothelial function assessed by reactive hyperemic index (RHI) in 68 Japanese volunteers. The authors observed that hydrogen water significantly increased RHI in these subjects. The topic that the authors investigated is clinically important. However, there are critical points that the authors should address.

Comments

1. The specific purpose of the study is vaguely stated. In addition, conclusion is also vaguely stated. The present study lacks a real atherosclerotic group with endothelial function (e.g., hypertensive patients, diabetes mellitus, dyslipidemia, and cardiovascular disease).

2. A number of studies in vitro reported that hydrogen directly works several types of cells through the various mechanisms. However, it is doubt that dissolved hydrogen in hydrogen water is taken in the vasculature and the cell after drinking hydrogen water in humans.

3. Subjects enrolled in this study were relatively young and healthy subjects. However, values of RHI were markedly small (e.g., 0.61 in the placebo group and 0.63 in the hydrogen water group). In addition, the authors stated that the border value of 0.71 seems to be close to that of the healthy masses (page 25, lines 6 and 7). This interpretation is clearly wrong. Previous studies including guidelines for management of endothelial function have clearly shown that normal range of RHI (healthy endothelial function) is ≥2.10, borderline is 1.67 to 2.10, and abnormal is <1.67.

4. Unfortunately, there are no mechanisms by which hydrogen water significantly increased RHI in healthy subjects. As the authors stated in the Discussion section, measurements of oxidative stress and inflammatory markers would draw more specific conclusions concerning the role of dissolved hydrogen in hydrogen water on vascular function.

5. It is well known that blood pressure, heart rate, glucose metabolism, and lipid profiles) other than BMI also are confounding factors for endothelial function. These parameters should be presented in Tables.

6. Introduction section and the Discussion section are too long. They should be significantly shortened as they contain a number of issues not directly related to the message of this study.

7. Figure 4 is too speculative and is not needed.

Reviewer #2: The manuscript addresses an interesting topic. The data are original and rich of information. The statistical methods employed are sound. Results are of potential interest. Some comments follow.

1. The data are not fully available. This is not in line with the journal's guidelines. More importantly, this does not allow the reviewer to check for the correctness and appropriatness of the methods. Please, add the dataset as a supplementary file.

2. The use of mixed effects models is sound. Linear mixed model could be an interesting option, but several aspects deserve more care. The model must fulfill some assumptions to ensure the reliability of the results. Those assumptions are completely overlooked. A complete analysis of the residuals must be included and tests (e.g log-likelihood ratio) should be included to check for the appropriateness of the random effects model. Moreover, some info on the random terms must be included (e.g. the variance of the random effects) and, as I am guessing that Gaussian distribution is taken from granted, if the assumptions on the random terms are also fulfilled.

3. Please, provide inference on the baseline characteristics as well, bearing in mind that any parametric statistical test is based on some assumptions to be fulfilled.

4. The way results are presented are somehow confusing to me. I would expect to see regression parameters, in which the placebo is taken as reference category. I do not really get how bot Placebo and Treatment can be included in a single model (as p-values are reported for both). Moreover, the term LS is not explained in the text. Please, report the results in a more common way, focusing on regression parameters and heterogeneity due to the longitudinal stracture in the data. Something should be said on the interaction effect as well.

Reviewer #3: The paper entitled “Peripheral endothelial function can be improved by daily consumption of water containing over 7ppm of dissolved hydrogen: A randomized controlled trial” describes the effect of oral intake of H2 improved the endothelial function assessed by reactive hyperemia peripheral arterial tonometry (RH-PAT).

In this study, subjects had an administration of 500mL of water every day. The subjects were divided into 2 groups by the property of water: high H2 water, and control. RH-PAT was performed at baseline, 1hour, 24 hours, and 14 days after the first intake. And the reactive hyperemia index (RHI), an index for a vascular endothelial function was compared between 2 groups. The authors found that the high H2 water group showed an increase in RHI in the later phase. And this effect was observed in subjects with lower RHI at baseline (lnRHI < 0.71). The authors have already reported this topic using flow mediated dilation (FMD), another index for endothelial dysfunction, and confirmed similar findings using another endothelial function test.

The paper suggests that oral intake of H2 might improve the endothelial function in healthy subjects having slightly reduced RHI. However, it is unclear if this treatment has a favorable effect on patients with impaired endothelial function.

Major comment

In general, the manuscript and too long and rambling. In Results, the same data were repeatedly shown in several figures and tables. This manuscript should be restyled with a more concise and clear way.

Minor comments

Introduction

- The introduction is too long, and describes too much for the comparison between FMD and EndoPAT.

- In the introduction (line 85), the authors stated: “the nature of the rosy faces often observed…..”. Do the authors have a reference for this?

Methods

- In line 121, “both 2d fingers” should read “both index fingers”.

- In this paper, RH-PAT was performed in the afternoon. The authors should describe the breakfast and other oral intakes like caffeine.

- The authors should describe the statistical analysis more precisely.

Results

- The authors divided subjects into low and high RHI groups with a cut-off of 0.71. They should explain how they conduced 0.71 as a cut-off.

Discussion

- In line 295-297, the authors stated: “The RH-PAT in the small vessels may be influenced by volume loading…”. Please show some references.

- In line 296-297, the authors said that loading of volume may strain the sympathetic nerve and affect PAT data. One of the advantages of RH-PAT is that RH-PAT might cancel the effect of systemic autonomic nervous tone by normalizing with contralateral PAT.

-

6. PLOS authors have the option to publish the peer review history of their article (what does this mean?). If published, this will include your full peer review and any attached files.

Reviewer #1: No

Reviewer #2: No

Reviewer #3: No

---

## [Author Response · Author response to Decision Letter 0]

1 Apr 2020

Responses to the Reviewers’ Comments

5. Review Comments to the Author

Reviewer #1: In this manuscript, Ishibashi et al. describe the effects of hydrogen water for 2 weeks on endothelial function assessed by reactive hyperemic index (RHI) in 68 Japanese volunteers. The authors observed that hydrogen water significantly increased RHI in these subjects. The topic that the authors investigated is clinically important. However, there are critical points that the authors should address.

Comments

Comment 1: The specific purpose of the study is vaguely stated. In addition, conclusion is also vaguely stated. The present study lacks a real atherosclerotic group with endothelial function (e.g., hypertensive patients, diabetes mellitus, dyslipidemia, and cardiovascular disease).

Response: Thank you very much for your suggestions.

The purposes of the present study were to assess the effects of H2 on endothelial function before morbidity and to provide the primary data of healthy people before applying H2 to a group with diseases related to endothelial dysfunction including atherosclerosis. Our study aims have been clearly stated in the Abstract section (page 2, lines 23-24) and Introduction section (page 5, lines 63-67; page 6, lines 79-83). The EndPad examination itself is the method used to prevent atherosclerosis before the subjects are involved in the group that already developed the diseases.

Therefore, the aims of the study were to investigate the effects of H2 on the endothelium of healthy populations and to determine whether H2 has beneficial effects in improving endothelial function before atherosclerosis and cardiovascular diseases develop. This is a basic study focusing on the primary effects of H2 on endothelial function of a healthy group, not people with morbidities, as noted in the text.

Page 2, lines 23-24_

“In this study, we aimed to investigate the effect of H2 on RH-PAT of the small arteries of fingers in healthy people.”

Page 5, lines 63-67

“It is important to not only estimate the function of the endothelium of both conduit arteries and peripheral arteries in healthy people using these non-invasive methodologies, but to also identify the factors or methodologies that can improve endothelial function and prevent endothelial dysfunction in healthy people without administering pharmaceutical drugs.”

Page 6, lines 76-80

“Besides these disease-conditions, it is important to identify safe and conventional ways for achieving relatively high (healthy) RHIs in the small vessels of healthy individuals.”

Additionally, one of the factors that made the Discussion section of our manuscript too long is the limitation of the insight where the effects of H2 is originated only for the scavenger property against hydroxyl radical. Many researchers and clinicians in this field are pointing to the limitations of the scavenger theory. Our study’s result cannot be explained by the diminishment of hydroxyl radicals and/or peroxynitrite by H2.

Recently, we proposed new mechanistic insights of H2 aside from the scavenger theory, and the first evidence to explain the insights was accepted at the end of 2019 after we submitted this manuscript and published in February 2020 [35]. We have shortened the Discussion and restructured it to make it concise and clear by adding explanations to the mechanisms by which H2 could improve the endothelial functions aside from the scavenger theory (page 22, line 311 to page 23, line 334).

Comment 2: A number of studies in vitro reported that hydrogen directly works several types of cells through the various mechanisms. However, it is doubt that dissolved hydrogen in hydrogen water is taken in the vasculature and the cell after drinking hydrogen water in humans.

Response: It has been clarified and agreed in this field that H2 is transferred into the human body and circulated from the ingested water containing H2. In cases where H2 was dissolved in water (15 ml/kg of 0.8-mM H2 water was administered to rats), H2 effectively alleviated nephrotoxicity caused by cisplatin at a concentration of 5-6 �M of H2 in blood [22]. In a clinical report about the concentration of H2 in blood 30 minutes after ingesting water containing H2, we also clarified that presence of hydrogen gas during exhalation from the lung of the person who digested the high H2 water can be explained by the rapid and passive diffusion of H2 in the aqueous human body including circulation [21].

We have added this explanation to the Methods section (page 9, line 140 to page 10, line 148).

[22] Nakashima-Kamimura N, Mori T, Ohsawa I, Asoh S, Ohta S. Molecular hydrogen alleviates nephrotoxicity induced by an anti-cancer drug cisplatin without compromising anti-tumor activity in mice. Cancer Chemother Pharmacol. 2009;64: 753-761.

Comment 3: Subjects enrolled in this study were relatively young and healthy subjects. However, values of RHI were markedly small (e.g., 0.61 in the placebo group and 0.63 in the hydrogen water group). In addition, the authors stated that the border value of 0.71 seems to be close to that of the healthy masses (page 25, lines 6 and 7). This interpretation is clearly wrong. Previous studies including guidelines for management of endothelial function have clearly shown that normal range of RHI (healthy endothelial function) is ≥2.10, borderline is 1.67 to 2.10, and abnormal is <1.67.

Response: The value you noted is not the natural logarithmic scaled RHI (Ln_RHI). Because RHI values are not normally distributed, RHI here is Ln_RHI, as described in the Methods section, [RHI=Ln{[RH-PAT ratio]×[0.226×Ln (baseline)–0.2]}[2,3, 20]. Therefore, most of the clinical trials using RHI including the large-scale clinicals trials usually adopt Ln_RHI of RHI, and some of them express the Ln_RHI simply as RHI.

We have added “and we used the natural logarithmic scaled Ln_RHI” in the Methods (page 8, lines 115-117) and changed “RHI” to “Ln_RHI” in the Abstract (page 2, lines 27, 30, 34, 35), Methods (page 10, lines 151, 152, 153, 157), Results (page 13, lines 182, 185, 195, 196; page 14, lines 199, 201, 203, 206; page 15, lines 211, 213; page 16, lines 215, 217, 219, 221, 223, 225, 227; page 17, lines 229, 231; page 18, line 234), and Discussion sections (page 18, lines 241, 243, 245, 247, 248, 249; page 19, lines 252, 258, 260, 261; page 20, lines 267, 269, 271; page 22, line 310)

Comment 4: Unfortunately, there are no mechanisms by which hydrogen water significantly increased RHI in healthy subjects. As the authors stated in the Discussion section, measurements of oxidative stress and inflammatory markers would draw more specific conclusions concerning the role of dissolved hydrogen in hydrogen water on vascular function.

Response: Recently, we proposed the new mechanistic insights of H2 aside from the scavenger theory, and the first evidence to explain the insights were accepted at the end of 2019 after we submitted this manuscript. We have shortened the Discussion and restructured it to make it concise and clear by adding the explanations for the mechanisms by which H2 can improve endothelial functions aside from the scavenger theory, as follows (page 22, line 311 to page 23, line 334).

“It should be noted that measuring RH of the peripheral microcirculation in which occlusion of the brachial artery following the release of blood flow is known as the model of ischemia reperfusion (I/R) injury, although the function of the vasodilation of the microvasculature is not fully understood [31]. The mechanisms of the cellular damage by I/R injury are believed to be caused by the superoxide burst in the mitochondrial respiratory chain. Particularly, the reverse electron transport accompanied by the elevated mitochondrial membrane potential is the major factor of electron leakage in mitochondrial complex I [32]. Recently, we have reported that H2 reduces the mitochondrial membrane potential in the living cultured cells [33]. In the study, the overproduction of superoxide with the high ratio of NADH/NAD+, which reflects the highly reduced state of mitochondria during hypoxia following I/R injury [34], was suppressed by H2. This situation corresponds to the right circle in Figure 4. On the basis of these most recent mechanistic insights of H2 [33, 35], we suppose that H2 may improve endothelial function via the mitochondrial mechanisms in healthy subjects. H2 may shift the circle in Figure 4 from right to left by reducing the elevated mitochondrial membrane potential as well as the overproduction of superoxide under the condition with hypoxia presented in the right circle.

Thus far, the limitations to explain the mechanisms of the H2 effect on a wide range of diseases or pathological states based on the scavenging property of H2 with respect to highly reactive radicals have been discussed [36, 37]. In blood flow, the availability of H2 and/or the frequency with which it reacts with target radicals seems insufficient to explain the distinct biological effects of H2 in the peripheral vasculature. In the present study, the continuous scavenging of deleterious radicals by H2 is excluded by the single intake of high H2 water in a day. Therefore, we consider that the effects of H2 herein are occur via the mitochondrial mechanisms described in Figure 4 [32-35].”

Comment 5: It is well known that blood pressure, heart rate, glucose metabolism, and lipid profiles) other than BMI also are confounding factors for endothelial function. These parameters should be presented in Tables.

Response: The alternations of blood pressure and heart rates have been added to Table 1.

We have added the comment below as a study limitation at the end of the Discussion section (page 24, lines 336-340).

As for glucose metabolism and lipid profiles, which are confounding factors for endothelial function, we did not collect blood samples, and our study protocol did not include the blood test.

Comment 6: Introduction section and the Discussion section are too long. They should be significantly shortened as they contain a number of issues not directly related to the message of this study.

Response: Thank you for your suggestion. We have deleted unnecessary sentences and restructured the Introduction section.

Introduction, page 5, line 71 to page 6, line 78

“In some interventional studies, attempts to improve the reactive hyperemia index (RHI) without using a pharmaceutical approach, but instead focusing on exercise therapy and/or lifestyle improvement, have been reported in patients suffering from heart failure and metabolic syndrome [15-18]. Improvements in RHI were also reported when postprandial hyperglycemia was treated pharmaceutically [19]. Besides these disease conditions, it is important to identify safe and conventional ways for achieving relatively high (healthy) RHIs in the small vessels of healthy individuals.”

Deleted sentences

“Although the evidence regarding the use of RH-PAT is lacking, the results obtained and reported after using this relatively new method are similar to those obtained for FMD, and the convenience and/or reproducibility of the methodologies are more universal in RH-PAT compared to that in FMD”

“This type of evaluation can elucidate the prognostic value of a new therapeutic modality, as well as determine the strategies for preventing any deleterious events associated with the use of this method.”

“So far, we have discussed the potential that H2 can react and dissociate to donate electrons to the highly active radical molecules such as �OH and ONOO-. According to guidelines on the safety of ingesting H2, therapeutic applications of H2 including clinical approaches have been emerging in the past decade.”

“Apart from the redox-related signalling including NO and the surrounding radical molecules such as H2O2, EDHF is attributed to electrochemical reactions by hyperpolarization of the endothelial cells first. The changes in the membrane potential of the endothelial cells can reach -85 〜 -80 mv, which is 10 〜 30 mv below the basal membrane potential of arteries and arterioles. This hyperpolarization is dependent on the activation of both endothelial Ca2+-sensitive K+-channels, the small KCa2.3 and KCa3.1. The efflux of K+ affecting the contiguous smooth muscle cells connected to the gap junctions is the primary factor in the mechanism of vasodilation of relatively small arteries. Because H2 is a stable molecule requiring 436 kJ/mol to cleave the covalent bond, and this property simultaneously permits the safety of H2 in our body, it is unlikely that the chemical reactions of H2 are directly involved in the mechanisms of EDHF.”

“In this study, healthy meant that no medications were required and/or not likely to become ill. In this study, however, quite a few subjects had RHI values below 0.71 despite their thinking that they were healthy (22 subjects in the placebo group and 19 subjects in the high H2 group).”

“…probably because of the single overload of the volume on the peripheral vasculature in the condition of low RH. The effects of the intake of 500 ml of water within 10 min is thought to have caused the increase in the volume of blood flow and affected the microcirculation independent of the presence of H2 or N2 in the water for individuals not accustomed to drinking such types of water. The RH-PAT in the small vessels may be influenced by volume loading more than by the diameter of the brachial artery as measured in FMD test. Furthermore, such loading of volume during the test may strain the sympathetic nerve and affect the PAT data. Within the high H2 group, an improvement was seen at 1 h after the drink in addition to at 24 h and 2 weeks after ingestion of the drink. This is consistent with the results we previously reported using FMD, where an H2-specific improvement was observed within 1 h after the consumption of 3.5 mg of H2. Currently, however, we could not assure the immediate efficacy of H2 on the peripheral artery, because of the presence of the placebo effect at 1 h in the low RHI group. Regarding the effect of H2 on RHI at 24 h, there was significant improvement from the RHI before consumption only within the H2 groups (Table 4), whereas in the low RHI groups, the difference from the value before ingestion was improved in the both of the placebo and H2 group. Further investigations about the immediate effects of a single ingestion of H2 within 24 h are necessary.”

Comment 7: Figure 4 is too speculative and is not needed.

Response: According to your suggestion, we have revised Figure 4 and deleting the schema below the circles. The remaining circles summarize the previous publications that are necessary for understanding our data here with the mitochondrial membrane potential, which is affected by H2, particularly for the researchers working on H2 medicine outside of the field of vasculature. We have also revised and shortened the Discussion section with regard to Figure 4, as described above (page 21, line 286 to page 23 line 334).

Reviewer #2: The manuscript addresses an interesting topic. The data are original and rich of information. The statistical methods employed are sound. Results are of potential interest. Some comments follow.

Comment 1: The data are not fully available. This is not in line with the journal's guidelines. More importantly, this does not allow the reviewer to check for the correctness and appropriatness of the methods. Please, add the dataset as a supplementary file.

Response: Thank you very much for your suggestions. We have added the dataset as Supplementary file S3.

Comment 2: The use of mixed effects models is sound. Linear mixed model could be an interesting option, but several aspects deserve more care. The model must fulfill some assumptions to ensure the reliability of the results. Those assumptions are completely overlooked. A complete analysis of the residuals must be included and tests (e.g log-likelihood ratio) should be included to check for the appropriateness of the random effects model. Moreover, some info on the random terms must be included (e.g. the variance of the random effects) and, as I am guessing that Gaussian distribution is taken from granted, if the assumptions on the random terms are also fulfilled.

Response: In the mixed effect models for repeated measures (MMRM), only correlation structure between outcome variables between time points is needed to be modeled explicitly [23]. Accordingly, the required assumption in this analysis is whether the correlation structure is correct. In this case, our analysis was unstructured; that is, we did not assume any structural assumptions for the correlation structure. MMRM is widely used for continuous outcome in longitudinal trials. We have described MMRM in the statistical analysis section (page 10, lines 153-155, 157-159).

[23] Mallinckrodt CH, Clark WS, David SR. Accounting for dropout bias using mixed-effects models. J Biopharm Stat. 2001;11: 9-21.

Comment 3: Please, provide inference on the baseline characteristics as well, bearing in mind that any parametric statistical test is based on some assumptions to be fulfilled.

Response: Statistical tests for baseline characteristics are not meaningful because these statistical strategies are not of interest in this trial, and a significance difference will occur with a probability of 0.05 [ref]. Therefore, we should interpret the difference based on a clinical viewpoint. In this study, there was no clinically meaningful difference in baseline characteristics between groups. In order to provide clarification, we have changed “The RHI value of the subjects was 0.61 ± 0.26 in the placebo group and 0.63 ± 0.29 in the high H2 group, and there was no significant difference.” to “Additionally, there was no significant difference in the RHI value of subjects between the groups (placebo group: 0.61±0.26, high H2 group: 0.63±0.29). We considered that the subjects in these two groups had physiologically similar characteristics in terms of endothelial function.” (page 11, lines 173-176).

[ref] Pocock SJ, Assmann SE, Enos LE, Kasten LE. Subgroup analysis, covariate adjustment and baseline comparisons in clinical trial reporting: current practice and problems. Stat Med. 2002;21: 2917-2130.

Comment 4: The way results are presented are somehow confusing to me. I would expect to see regression parameters, in which the placebo is taken as reference category. I do not really get how bot Placebo and Treatment can be included in a single model (as p-values are reported for both). Moreover, the term LS is not explained in the text. Please, report the results in a more common way, focusing on regression parameters and heterogeneity due to the longitudinal stracture in the data. Something should be said on the interaction effect as well.

Response: We have revised Tables 3-6 and summarized the data in Tables 2 and 3 for simplicity. The MMRM included the treatment group, time points, and interaction between the treatment group and the time points as fixed effects, with the RHI at baseline as a covariate. Therefore, we were able to estimate the covariate-adjusted mean and mean difference by estimated correlation coefficients. Additionally, the main purpose of this trial was to compare the adjusted mean between the groups, so we did not show many estimated parameters. LS is the covariate-adjusted mean, and we have revised the following sentence in the Results section to provide clarification (page 11, lines 173-176).

 “Additionally, there was no significant difference in the RHI value of subjects between the groups (placebo group: 0.61±0.26, high H2 group: 0.63±0.29). We considered that the subjects in these two groups had physiologically similar characteristics in terms of endothelial function.”

Reviewer #3: The paper entitled “Peripheral endothelial function can be improved by daily consumption of water containing over 7ppm of dissolved hydrogen: A randomized controlled trial” describes the effect of oral intake of H2 improved the endothelial function assessed by reactive hyperemia peripheral arterial tonometry (RH-PAT).

In this study, subjects had an administration of 500mL of water every day. The subjects were divided into 2 groups by the property of water: high H2 water, and control. RH-PAT was performed at baseline, 1hour, 24 hours, and 14 days after the first intake. And the reactive hyperemia index (RHI), an index for a vascular endothelial function was compared between 2 groups. The authors found that the high H2 water group showed an increase in RHI in the later phase. And this effect was observed in subjects with lower RHI at baseline (lnRHI < 0.71). The authors have already reported this topic using flow mediated dilation (FMD), another index for endothelial dysfunction, and confirmed similar findings using another endothelial function test.

The paper suggests that oral intake of H2 might improve the endothelial function in healthy subjects having slightly reduced RHI. However, it is unclear if this treatment has a favorable effect on patients with impaired endothelial function.

Major comment

Comment 1: In general, the manuscript and too long and rambling. In Results, the same data were repeatedly shown in several figures and tables. This manuscript should be restyled with a more concise and clear way.

Response: Thank you very much for your suggestions. One of the factors that made the Discussion section of our manuscript too long is the limitation of the insight where the effects of H2 are originated from only the scavenger property against hydroxyl radical. Many researchers and clinicians in this field are pointing out the limitations of the scavenger theory. Our study’s results cannot be explained by the diminishment of hydroxyl radicals and/or peroxynitrite by H2.

Recently, we proposed new mechanistic insights of H2 aside from the scavenger theory, and the first evidence to explain the insights were accepted at the end of 2019 after we submitted this manuscript and published in February 2020 [35]. We have shortened the Discussion and restructured it to make it concise and clear by adding explanations for the mechanisms by which H2 can improve endothelial functions aside from the scavenger theory.

Further, we have deleted unnecessary sentences and revised text in the Introduction section.

Introduction: page 5, line 71 to page 6, line 78

“In some interventional studies, attempts to improve the reactive hyperemia index (RHI) without using a pharmaceutical approach, but instead focusing on exercise therapy and/or lifestyle improvement, have been reported in patients suffering from heart failure and metabolic syndrome. Improvements in RHI were also reported when postprandial hyperglycemia was treated pharmaceutically. Besides these disease conditions, it is important to identify safe and conventional ways for achieving relatively high (healthy) RHIs in the small vessels of healthy individuals.”

Deleted sentences

“Although the evidence regarding the use of RH-PAT is lacking, the results obtained and reported after using this relatively new method are similar to those obtained for FMD, and the convenience and/or reproducibility of the methodologies are more universal in RH-PAT compared to that in FMD

“This type of evaluation can elucidate the prognostic value of a new therapeutic modality, as well as determine the strategies for preventing any deleterious events associated with the use of this method.”

We have also restructured the Results section to shorten it and focus on the RHI values. We have revised Table 1 to clearly present the basic data of the subjects by combining the baseline data in Table 2, and we renumbered the tables. Data in Tables 2-5 include the same data of Figures 2 and 3, as you pointed out. We deleted the column for “Observed Value” from Tables 2-5. Although we considered whether Table 2-5 were essential, we considered that they are necessary for the statistical considerations that reviewer #2 requested. We have combined the data in Tables 2 and 3 into Table 2, and combined the data in Tables 4 and 5 into Table 3 in order to simplify the presentation for the statistical data.

We have deleted the following sentences from the Discussion section in order to shorten and restructure it.

“So far, we have discussed the potential that H2 can react and dissociate to donate electrons to the highly active radical molecules such as �OH and ONOO-. According to guidelines on the safety of ingesting H2, therapeutic applications of H2 including clinical approaches have been emerging in the past decade.”

“Apart from the redox-related signaling including NO and the surrounding radical molecules such as H2O2, EDHF is attributed to electrochemical reactions by hyperpolarization of the endothelial cells first. The changes in the membrane potential of the endothelial cells can reach -85 ~ -80 mv, which is 10 ~ 30 mv below the basal membrane potential of arteries and arterioles. This hyperpolarization is dependent on the activation of both endothelial Ca2+-sensitive K+-channels, the small KCa2.3 and KCa3.1. The efflux of K+ affecting the contiguous smooth muscle cells connected to the gap junctions is the primary factor in the mechanism of vasodilation of relatively small arteries. Because H2 is a stable molecule requiring 436 kJ/mol to cleave the covalent bond, and this property simultaneously permits the safety of H2 in our body, it is unlikely that the chemical reactions of H2 are directly involved in the mechanisms of EDHF.”

“In this study, healthy meant that no medications were required and/or not likely to become ill. In this study, however, quite a few subjects had RHI values below 0.71 despite their thinking that they were healthy (22 subjects in the placebo group and 19 subjects in the high H2 group).”

“The RH-PAT in the small vessels may be influenced by volume loading more than by the diameter of the brachial artery as measured in FMD test. Furthermore, such loading of volume during the test may strain the sympathetic nerve and affect the PAT data. Within the high H2 group, an improvement was seen at 1 h after the drink in addition to at 24 h and 2 weeks after ingestion of the drink. This is consistent with the results we previously reported using FMD, where an H2-specific improvement was observed within 1 h after the consumption of 3.5 mg of H2. Currently, however, we could not assure the immediate efficacy of H2 on the peripheral artery, because of the presence of the placebo effect at 1 h in the low RHI group. Regarding the effect of H2 on RHI at 24 h, there was significant improvement from the RHI before consumption only within the H2 groups (Table 4), whereas in the low RHI groups, the difference from the value before ingestion was improved in the both of the placebo and H2 group. Further investigations about the immediate effects of a single ingestion of H2 within 24 h are necessary.”

Minor comments

Introduction

Comment 2: The introduction is too long, and describes too much for the comparison between FMD and EndoPAT.

Response: We have deleted sentences from the Introduction section to shorten and restructure it, as noted above in our response to your comment 1.

We have added the following sentences in the Discussion section (page 22, line 311 to page 23, line 334), according to the recent finding about the new properties of H2.

“It should be noted that measuring RH of the peripheral microcirculation in which occlusion of the brachial artery following the release of blood flow is known as the model of ischemia reperfusion (I/R) injury, although the function of the vasodilation of the microvasculature is not fully understood [29]. The mechanisms of the cellular damage by I/R injury are believed to be caused by the superoxide burst in the mitochondrial respiratory chain. Particularly, the reverse electron transport accompanied by the elevated mitochondrial membrane potential is the major factor of electron leakage in mitochondrial complex I [30]. Recently, we have reported that H2 reduces the mitochondrial membrane potential in the living cultured cells [31]. In the study, the overproduction of superoxide with the high ratio of NADH/NAD+, which reflects the highly reduced state of mitochondria during hypoxia following I/R injury [32], was suppressed by H2. This situation corresponds to the right circle in Figure 4. On the basis of these most recent mechanistic insights of H2 [31, 33], we suppose that H2 may improve endothelial function via the mitochondrial mechanisms in healthy subjects. H2 may shift the circle in Figure 4 from right to left by reducing the elevated mitochondrial membrane potential as well as the overproduction of superoxide under the condition with hypoxia presented in the right circle.

Thus far, the limitations to explain the mechanisms of the H2 effect on a wide range of diseases or pathological states based on the scavenging property of H2 with respect to highly reactive radicals have been discussed [34, 35]. In blood flow, the availability of H2 and/or the frequency with which it reacts with target radicals seems insufficient to explain the distinct biological effects of H2 in the peripheral vasculature. In the present study, the continuous scavenging of deleterious radicals by H2 is excluded by the single intake of high H2 water in a day. Therefore, we consider that the effects of H2 herein are occur via the mitochondrial mechanisms described in Figure 4 [30-33].”

Comment 3: In the introduction (line 85), the authors stated: “the nature of the rosy faces often observed….”. Do the authors have a reference for this?

Response: No, it is just a consensus among the clinicians treating H2; we do not have a reference to support this statement. We have deleted it.

Methods

Comment 4: In line 121, “both 2d fingers” should read “both index fingers”.

Response: Thank you very much for your advice. We have revised the text accordingly (page 7, line 105).

Comment 5: In this paper, RH-PAT was performed in the afternoon. The authors should describe the breakfast and other oral intakes like caffeine.

Response: We have described the 6 h fasting as follows (page 7, lines 95-96): “All subjects were asked to fast and avoid drinking caffeinated drinks or ingesting sugar for 6 h before the test.” We have also described the breakfast (page 7, line 96): “All subjects were asked to fast and avoid drinking caffeinated drinks or ingesting sugar as well as breakfast for 6 h before the test.”

Comment 6: The authors should describe the statistical analysis more precisely.

Response: We responded to the queries and requests of reviewer #2 about the statistical analysis, and we described MMRM (page 10, lines 153-155, 157-159).

Results

Comment 7: The authors divided subjects into low and high RHI groups with a cut-off of 0.71. They should explain how they conduced 0.71 as a cut-off.

Response: We explained the cut-off in the Discussion section, and we have revised the sentence and added a comment in the Results section.

Results: page 16, lines 215-217

“In order to identify the influence of the RHI value before ingestion on the data at each time point, we divided the subjects into two groups with the cut-off value of 0.71 (see Discussion section).”

Discussion: page 18, line 247 to page 19, line 259

“Although not intended for healthy subjects, there are several investigations on the prognostic value of RHI for predicting cardiovascular risks. The RHI values reported were 0.531 in 577 patients at high risk for cardiovascular events by Matsuzawa et al [6] and 0.4 in 270 low-risk outpatients with chest pain by Rubinshtein et al [7]. It has been also reported that the risk of arrhythmia recurrence could be estimated by an RHI value of 0.618 in 92 participants with atrial fibrillation undergoing catheter ablation [23]. Orthopedic patients who underwent total hip or knee arthroplasty and avoided postoperative deep vein thrombosis (DVT) had preoperative ln-RHI values of 0.71 ± 0.25 [24]. Among these reports, the border value of 0.71 for the orthopedic patients avoiding DVT seems to be close to that of the healthy masses. To predict future endothelial health in healthy subjects in this study, it seemed to be appropriate to set the threshold for RHI above the values for patients with cardiovascular risks. We set the RHI value at 0.71 in order to divide the subjects into two groups.”

Discussion

Comment 8: In line 295-297, the authors stated: “The RH-PAT in the small vessels may be influenced by volume loading…”. Please show some references.

Response: Thank you for pointing out this reference. Because the comments concerning the placebo effects were based on our speculative opinion, we searched for appropriate references. Unfortunately, we could not find a reference that clearly explained the relationship between hyperemia of peripheral circulation and intake of 500 ml of water within 1 h. Therefore, we deleted the explanation of the placebo effect attributing to overload of the volume on the peripheral vasculature in the condition of low RH. Instead, we stated (page 19, line 266 to page 20, line 268), “…although it is currently unclear whether it is merely the placebo effect or there is another factor that causes the improvement of RHI within 1 h after the placebo group drinks the water containing N2.”

Comment 9: In line 296-297, the authors said that loading of volume may strain the sympathetic nerve and affect PAT data. One of the advantages of RH-PAT is that RH-PAT might cancel the effect of systemic autonomic nervous tone by normalizing with contralateral PAT.

Response: Thank you very much for pointing this out. This comment is in contradiction with the theory of the PAT test so we deleted it. We also deleted the unnecessary sentence explaining the effects at 1 h compared with the results of the FMD test.

6. PLOS authors have the option to publish the peer review history of their article (what does this mean?). If published, this will include your full peer review and any attached files.

Do you want your identity to be public for this peer review? For information about this choice, including consent withdrawal, please see our Privacy Policy.

Reviewer #1: No

Reviewer #2: No

Reviewer #3: No

---

## [Decision Letter · Decision Letter 1]

17 Apr 2020

PONE-D-19-26280R1

Peripheral endothelial function can be improved by daily consumption of water containing over 7 ppm of dissolved hydrogen: A randomized controlled trial

PLOS ONE

Dear Dr. Ishibashi,

Thank you for submitting your manuscript to PLOS ONE. After careful consideration, we feel that it has merit but does not fully meet PLOS ONE’s publication criteria as it currently stands. Therefore, we invite you to submit a revised version of the manuscript that addresses the points raised during the review process.

Statistical reviewer still asks to verify assumption of the modelling while another reviewer still underscores the lack of a control group.

We would appreciate receiving your revised manuscript by Jun 01 2020 11:59PM. To enhance the reproducibility of your results, we recommend that if applicable you deposit your laboratory protocols in protocols.io, where a protocol can be assigned its own identifier (DOI) such that it can be cited independently in the future. For instructions see: http://journals.plos.org/plosone/s/submission-guidelines#loc-laboratory-protocols

We look forward to receiving your revised manuscript.

Kind regards,

Giuseppe Andò, M.D., Ph.D.

Academic Editor

PLOS ONE

Reviewers' comments:

Reviewer's Responses to Questions

**Comments to the Author**

1. If the authors have adequately addressed your comments raised in a previous round of review and you feel that this manuscript is now acceptable for publication, you may indicate that here to bypass the “Comments to the Author” section, enter your conflict of interest statement in the “Confidential to Editor” section, and submit your "Accept" recommendation.

Reviewer #1: (No Response)

Reviewer #2: (No Response)

Reviewer #3: All comments have been addressed

2. Is the manuscript technically sound, and do the data support the conclusions?

Reviewer #1: Partly

Reviewer #2: Partly

Reviewer #3: Yes

3. Has the statistical analysis been performed appropriately and rigorously? 

Reviewer #1: No

Reviewer #2: Yes

Reviewer #3: Yes

4. Have the authors made all data underlying the findings in their manuscript fully available?

Reviewer #1: (No Response)

Reviewer #2: Yes

Reviewer #3: Yes

5. Is the manuscript presented in an intelligible fashion and written in standard English?

Reviewer #1: Yes

Reviewer #2: Yes

Reviewer #3: Yes

6. Review Comments to the Author

Reviewer #1: The present study lacks a real atherosclerotic group with endothelial function (e.g., hypertensive patients, diabetes mellitus, dyslipidemia, and cardiovascular disease).

A number of studies in vitro reported that hydrogen directly works several types of cells through the various mechanisms. However, it is doubt that dissolved hydrogen in hydrogen water is taken in the vasculature and the cell after drinking hydrogen water in humans.

Reviewer #2: I appreciate most the reply given by the authors to my comments.

There are still some points deserving a better clarification.

Previously, I commented that "A complete analysis of the residuals must be included and

tests (e.g log-likelihood ratio) should be included to check for the appropriateness of

the random effects model. Moreover, some info on the random terms must be included

(e.g. the variance of the random effects) and, as I am guessing that Gaussian

distribution is taken from granted, if the assumptions on the random terms are also

fulfilled.". The authors reply is not satisfactory. The unstructured specification mentioned in their reply refers to the variance-covariance matrix of the random effects. This is quite a flexible assumption and I am fine with it.

The issue is that the random effects follow some distribution. Again I guess it is the Gaussian one, but this must be checked somehow, as departure from the Gaussian assumption may lead to biased estimates. Moreover, as the author replied, random effects modelling is widely used for continuous outcome in longitudinal trials. This is absolutely true. But continuous outcomes may be distributed according to hundreds of different distributions. Again, the most used one is the Gaussian, i.e. the error term in the linear predictor is assumed to follow a zero-mean Gaussian distribution. This is a crucial assumption of the modelling. If heavy tails are evident, i.e. if the error term follows a t distribution instead, regression parameters and all the statistical inference are biased. This is the reason why the analysis of residuals must be provided, because the linear mixed model is based on some assumptions. If those assumptions are not fulfilled, the results are unreliable. Citing a paper in the J. Bioph. Stat is not a good choice, as dozens of statistical books exist (just to mention one https://www.routledge.com/Longitudinal-Data-Analysis/Fitzmaurice-Davidian-Verbeke-Molenberghs/p/book/9781584886587) and discuss longitudinal data analysis, mentioning how to perform a proper data analysis.

Reviewer #3: The authors correctly addressed all the points raised by reviewers, and the revised manuscript is much improved.

7. PLOS authors have the option to publish the peer review history of their article (what does this mean?). If published, this will include your full peer review and any attached files.

Reviewer #1: No

Reviewer #2: No

Reviewer #3: No

---

## [Author Response · Author response to Decision Letter 1]

4 May 2020

Peripheral endothelial function can be improved by daily consumption of water containing over 7 ppm of dissolved hydrogen: A randomized controlled trial (Manuscript: PONE-D-19-26280R1)

Response to Reviewer #1

Thank you for reviewing our manuscript and for your insightful comments. We have responded to each of your comments below, and have revised the manuscript accordingly.

Comment 1: The present study lacks a real atherosclerotic group with endothelial function (e.g., hypertensive patients, diabetes mellitus, dyslipidemia, and cardiovascular disease).

Response: Thank you for your further suggestions.

In the former round of revision, we emphasized the aim and goal of the present study according to your previous suggestion in the Abstract section:

“In this study, we aimed to investigate the effect of H2 on RH-PAT of the small arteries of fingers in healthy people.”

…and in the Introduction section:

“It is important to not only estimate the function of the endothelium of both conduit arteries and peripheral arteries in healthy people using these non-invasive methodologies, but to also identify the factors or methodologies that can improve endothelial function and prevent endothelial dysfunction in healthy people without administering pharmaceutical drugs.”

“Besides these disease-conditions, it is important to identify safe and conventional ways for achieving relatively high (healthy) RHIs in the small vessels of healthy individuals.”

Furthermore, we added the comments in the Study limitations as follows (page 27, lines 342-347):

“As the aim of this study was to investigate the effect of H2 on RH-PAT intended for healthy subjects to assess the significance of daily consumption of H2 to prevent the endothelial dysfunction of small arteries, data for an atherosclerotic group with a low RHI value are lacking. It is important to examine the effect of H2 on RH-PAT of people having a high risk of hypertension, cardiovascular diseases, diabetes mellitus and dyslipidemia due to atherosclerosis. The next study should include participants with these characteristics.”

Comment 2: A number of studies in vitro reported that hydrogen directly works several types of cells through the various mechanisms. However, it is doubt that dissolved hydrogen in hydrogen water is taken in the vasculature and the cell after drinking hydrogen water in humans.

Response: As we explained in the Methods section (page 9, lines 138-140), it has been clarified and agreed in this field that H2 is transferred into the human body and the hydrogen is absorbed in vasculature and circulated from ingested water containing H2. (reference [22]). 

Moreover, we added reference [23] and revised the sentence as follows (pages 9-10, lines 140-142):

“In cases where rats were administered 15 mL/kg of 0.8 mM H2 water by a catheter or mice drank approximately 200mL/kg of 0.2–0.4 mM H2 water per day, the concentration of H2 in their blood was 5–6 �M and about 2 �M, respectively [22, 23].” 

Response to Reviewer #2

Thank you for reviewing our manuscript and for your insightful comments. We have responded to each of your comments below, and have revised the manuscript accordingly.

Comment: I appreciate most the reply given by the authors to my comments.

There are still some points deserving a better clarification.

Previously, I commented that "A complete analysis of the residuals must be included and tests (e.g log-likelihood ratio) should be included to check for the appropriateness of the random effects model. Moreover, some info on the random terms must be included (e.g. the variance of the random effects) and, as I am guessing that Gaussian distribution is taken from granted, if the assumptions on the random terms are also fulfilled.". The authors reply is not satisfactory. The unstructured specification mentioned in their reply refers to the variance-covariance matrix of the random effects. This is quite a flexible assumption and I am fine with it. The issue is that the random effects follow some distribution. Again I guess it is the Gaussian one, but this must be checked somehow, as departure from the Gaussian assumption may lead to biased estimates. Moreover, as the author replied, random effects modelling is widely used for continuous outcome in longitudinal trials. This is absolutely true. But continuous outcomes may be distributed according to hundreds of different distributions. Again, the most used one is the Gaussian, i.e. the error term in the linear predictor is assumed to follow a zero-mean Gaussian distribution. This is a crucial assumption of the modelling. If heavy tails are evident, i.e. if the error term follows a t distribution instead, regression parameters and all the statistical inference are biased. This is the reason why the analysis of residuals must be provided, because the linear mixed model is based on some assumptions. If those assumptions are not fulfilled, the results are unreliable. Citing a paper in the J. Bioph. Stat is not a good choice, as dozens of statistical books exist (just to mention one https://www.routledge.com/Longitudinal-Data-Analysis/Fitzmaurice-Davidian-Verbeke-Molenberghs/p/book/9781584886587) and discuss longitudinal data analysis, mentioning how to perform a proper data analysis.

Response: Thank you very much for your detailed advice. As you guessed, we assumed a Gaussian distribution for the error term in MMRM. I checked distribution of the residuals in the MMRM model for Table 2 and Table 3. Please see attached figure below. The distribution appears to be normal/ Gaussian (upper right) and the points approximately fall on the line in QQ-plot (lower left). In addition, we performed a log-likelihood ratio test in order to check the fit of the MMRM model to the data and its p-values were 0.0483 and 0.3597 for Table 2 and Table 3, respectively. The p-value of Table 3 (0.3597) is relatively large because of the small sample size. We think that the MMRM model is appropriate for the data. We have added a related sentence to the Statistical analysis section (pages 10-11, lines 157-159, and footnotes to Table 2 (page 17, line 215) and Table 3 (pages 20, line 237) as follows:

(Page 10-11, lines 157-159)

“Furthermore, we assumed a Gaussian error term in the model. We performed a log-likelihood ratio test and residual analysis in order to check the fit of the model.”

(Page 17, line 215)

“p-value of log-likelihood test: 0.0483”

(Page 20, line 237)

“p-value of log-likelihood test: 0.3597”

Furthermore, thank you for providing us with an appropriate reference. We replaced reference [24] according to your suggestion as follows:

[24] Garrett Fitzmaurice, Marie D, Geert V, Geert M. (2008). Longitudinal data analysis. Chapman and Hall/CRC. 

Residual analysis for Table 1

Residual analysis for Table 2 (Figure in uploaded file)

---

## [Editor Report · Decision Letter 2]

7 May 2020

Peripheral endothelial function can be improved by daily consumption of water containing over 7 ppm of dissolved hydrogen: A randomized controlled trial

PONE-D-19-26280R2

Dear Dr. Ishibashi,

We are pleased to inform you that your manuscript has been judged scientifically suitable for publication and will be formally accepted for publication once it complies with all outstanding technical requirements.

With kind regards,

Giuseppe Andò, M.D., Ph.D.

Academic Editor

PLOS ONE
---

## [Editor Report · Acceptance letter]

20 May 2020

PONE-D-19-26280R2 

Peripheral endothelial function can be improved by daily consumption of water containing over 7 ppm of dissolved hydrogen: A randomized controlled trial 

Dear Dr. Ishibashi:

I am pleased to inform you that your manuscript has been deemed suitable for publication in PLOS ONE. Congratulations! Your manuscript is now with our production department. 

With kind regards,

on behalf of

Dr. Giuseppe Andò 

Academic Editor

PLOS ONE